# Business-as-usual and fantasy planning – an analysis of equity within climate adaptation planning for sanitation in Nairobi

Leonie K. Hyde-Smith[1], Daniel Ddiba[2], Sarah Dickin[3], Domenic Kiogora[4], Anna L. Mdee[5], Vanessa Reinfelder[6], Katy E. Roelich[7], Barbara Evans[1]*

1 School of Civil Engineering, University of Leeds, Leeds, United Kingdom, 2 Stockholm Environment Institute, Stockholm, Sweden, 3 Swedesd - Sustainability Learning and Research Centre, Department of Women's and Children's Health, Uppsala University, Sweden, 4 Meru University of Science and Technology, Department of Biological Sciences, Meru, Kenya, 5 School of Politics and International Studies, University of Leeds, Leeds, United Kingdom, 6 gruppeF, Berlin, Germany, 7 School of Earth and Environment/ Sustainability Research Institute, University of Leeds, Leeds, United Kingdom

* B.E.Evans@leeds.ac.uk

## Abstract

This paper explores the disconnect between policy rhetoric and implementation at the intersection of sanitation equity and climate change in Nairobi, Kenya. To examine the current sanitation adaptation trajectory, we reviewed Nairobi's sanitation policies, planning, and investment frameworks, focusing on their integration with climate adaptation strategies and consideration of equity in terms of distribution, recognition and processes. We used a socio-technical regime framework to map the current sanitation service configurations in Nairobi and projected their future under different climate change scenarios. Our findings provide evidence for a disconnection between policy rhetoric and implementation, prioritising sewerage development at the expense of other sanitation regimes. Despite recognising equity issues in policy, substantive action towards equitable sanitation governance is lacking. This imbalance hinders the realisation of the constitutionally recognised right to adequate sanitation in the foreseeable future. The anticipated impact of climate change on Nairobi's sanitation sector suggests an exacerbation of existing service inequalities. Our projection indicates that by 2030, a sizeable portion of Nairobi's residents will experience poor sanitation services. Our study emphasizes the critical need for a fundamental paradigm shift. It calls for a robust and honest discussion on delivering high-quality, resilient sanitation services at scale including both sewer and non-sewered sanitation and necessitating substantial public investment and support for all systems. This reappraisal is imperative for ensuring equitable and sustainable sanitation solutions in the face of climate change.

**Data availability statement:** All relevant data are within the manuscript and its Supporting Information files.

**Funding:** This work was solely supported by the UKRI Engineering and Physical Science Research Council (EPSRC). EPSRC Grant number: EP/S022066/1. There were no other sources of funding.

**Competing interests:** The authors have declared that no competing interests exist.

## Background

The 6th Intergovernmental Panel on Climate Change (IPCC) Assessment Report has highlighted that climate change amplifies inequalities and undermines sustainable development across regions [1, p.1174]. Climate change-driven hazards are poised to (adversely) affect urban sanitation systems across the globe [2], requiring adaptation efforts by governments, local authorities, service providers, and households.

Meanwhile, inequitable access to sanitation service provision is still prevalent globally [3–5] and gained heightened attention during the Covid-19 response [6–8]. However, sanitation investments and funding arrangements are biased towards subsidising capital investment and operational costs of centralised sewerage and treatment systems, which often primarily tend to more affluent parts of the population [9–12]. These systemic biases create what we describe as a *business-as-usual* approach, where the focus on traditional, centralised systems perpetuates inequalities and limits progress towards universal sanitation access.

There is growing recognition that climate-adaptive sanitation infrastructure is non-negotiable for avoiding the risk of reversing progress towards universal access to safe sanitation services [13,14] and thereby worsening inequalities. However, attention to adaptation needs within urban sanitation systems has been limited and primarily focused on technology solutions [15]. Sanitation adaptation is mainly happening in high-income countries, focusing on retrofitting and upgrading old infrastructure approaching the end of its design life [16,17].

Equity considerations in climate change adaptation efforts are generally assessed through a justice lens [18]. The commonly adopted conceptualisations of climate (adaptation) justice are often derived from the environmental justice scholarship [19,20] and are frequently composed of multiple components such as distributive, procedural and recognition justice [21–23]. A global systematic review of peer-reviewed empirical research on adaptation responses [18] found that around 60% of the 1,682 reviewed adaptation responses considered social equity in terms of distributive and procedural justice aspects. However, unlike the more advanced scholarship on just transitions for other urban sectors such as energy [21,22,24,25] or mobility [26,27] justice aspects on the interface between urban sanitation planning and climate change adaptation are relatively underexplored.

Against this backdrop, this study explores if and how sanitation equity aspects are considered in sanitation adaptation planning in a city with an existing sanitation service gap and service inequalities.

As explored by various scholars (e.g., [28, 29, 30]), the concept of socio-technical regimes provides an integrated perspective on the interaction between social practices, institutions, and technologies within a sector. Urban sanitation transcends mere infrastructure. In the context of the sanitation sector, socio-technical regimes offer a nuanced understanding of how sanitation systems are designed, maintained, and adapted over time, incorporating both technical and social dimensions. We selected the example of Nairobi, Kenya, since there is good data availability on the sanitation sector of Nairobi [31–33] and Kenya [34]. Previous analyses by Van Welie et al.

[31] and Mdee et al [30] have examined Nairobi's sanitation sector using a socio-technical regime framework providing a strong foundation for further exploration. Nairobi exemplifies extreme sanitation inequalities rooted in the city's colonial history and which have led a highly fragmented or 'splintered' sanitation sector [30,31,35]. Nairobi's sanitation sectoral regime is marked by heterogeneity and disjointed coexisting service regimes of different grades of formality particularly in non-sewered areas, [30,33,36]. Sanitation governance responsibilities are dispersed across multiple state departments, while informal settlements are rendered 'illegal' placing them beyond the scope of state accountability [30].

The distinctive configuration of Nairobi's sanitation services is the result of its unique historical, cultural, and economic contexts [35,37–39], necessitating a nuanced understanding of these factors. However, there are limited studies on the equity of water and sanitation services in Nairobi [31,32,40]. Throughout the last two decades, Kenya's water sector has (at least rhetorically) proclaimed the intention to engage with low-income groups through a 'pro-poor' or 'inclusive' service approach. While there have been numerous applaudable and innovative sanitation efforts in Nairobi reflecting this commitment, these initiatives have struggled to achieve the scale necessary to effectively address the needs of marginalized populations. The Kenyan Constitution recognises the human right to sanitation, although there is little evidence that formal inclusion in the constitution has led to improved service delivery [41]. Furthermore, the 2023 hosting of the inaugural African Climate Summit in Nairobi underscores a strong political extraversion to signal to international audiences the dedication to climate change adaptation and mitigation policy, even though our study points to discrepancies between these proclaimed commitments and actual implementation.

This sudy aims to examine the disparities between proclaimed – *fantasy plans* – and actual equity considerations at the intersection of urban sanitation planning and climate change adaptation in a city characterised by substanital sanitation inequalities in a resource constraint setting.

Our analysis follows several objectives:

1. Characterise the current sanitation service configuration in Nairobi and inherent sanitation service inequalities from a distributive justice perspective

2. Evaluate how the different sanitation service configurations are likely to be affected by climate change hazards

3. Evaluate the adequacy of Nairobi's policy and planning frameworks in addressing the additional pressures posed by climate change on sanitation systems. This involves assessing whether and how climate adaptation is incorporated into the city's sanitation planning and investment framework, and conversely, whether sanitation considerations are integrated into Nairobi's broader climate adaptation planning and investment framework. The aim is to determine whether these frameworks are fit-for-purpose to prevent increased vulnerabilities and inequalities resulting from potential climate hazard related sanitation system failures

4. Estimate how the effects of climate change are likely to alter service inequalities in Nairobi, considering the current sanitation adaptation planning framework.

## Methods

### Visualisation and quantification of current sanitation service configurations in Nairobi

We examined the unequal distribution of sanitation services across Nairobi, employing a socio-technical regime categorisation to map of sanitation services (Fig 1).

To understand the current layout of sanitation services in Nairobi County, we used georeferenced data (shapefiles showing land use and income classes) from previous studies on water inequality in Nairobi [40,42] and cross-referenced these with recent satellite imagery [43] to verify consistency and detect any changes in land use or income classification patterns. Key indicators, such as visible infrastructure developments, road networks, and urban expansion, were cross-checked to

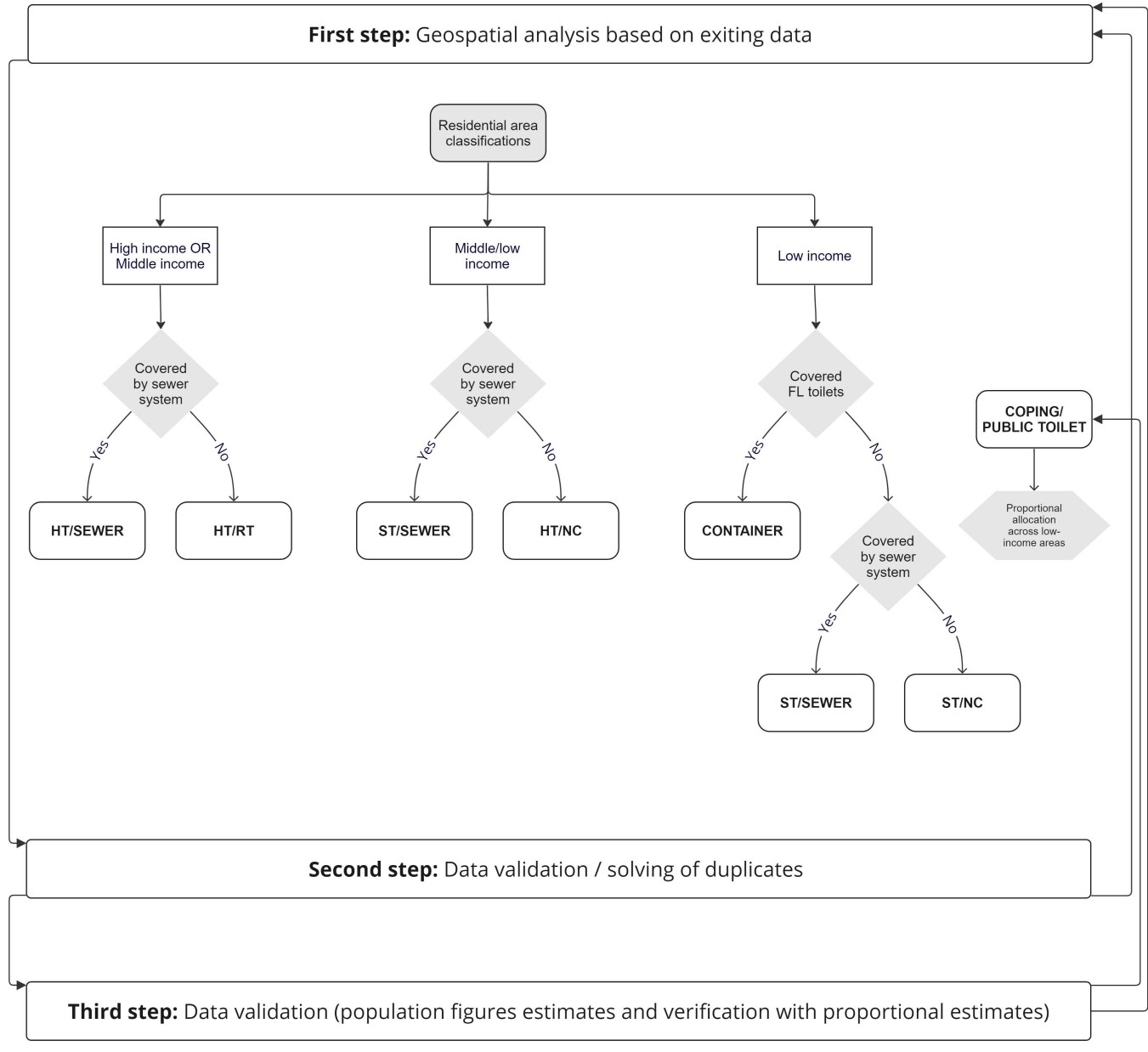

COPING (NC/NT) = People don't have access to toilets us coping mechanisms; PUBLIC TOILET = People rely on public toilets as main form of sanitation; ST/NC = Shared toilets that are NOT adequately managed; CONTAINER (ST/RT) = Shared toilets that are regularly emptied; ST/SEWER = Shared toilets connected to the sewer system; HT/NC = Household that are not adequately managed; HT/RT = Household toilets that regularly emptied; HT/SEWER = Household toilets connected to the sewer system

**Fig 1. Description of sanitation regime categorisations and mapping process (current situation).**

ensure the shapefiles matched current urban layouts. Given the rapid urbanisation and population growth in Nairobi [40,44], the most recent classifications of income-specific residential and non-residential areas [40] were used.

We overlaid a map of the current and planned sewer network from the Nairobi Masterplan and combined this data with past analysis of sanitation coverage and residential typology [42], alongside current information on sanitation gathered from various documents and stakeholders in Nairobi. Comparing this data with the recent basemap, we conducted a geospatial analysis to map the main types of sanitation systems in Nairobi following the allocation process described in Fig 1.

To estimate the number of people served by each type of sanitation system, we overlaid the shapefiles with updated gridded population data for 2022 [45] produced by the WorldPop Research Group at the University of Southampton. WorldPop derive population data from projections using the 2019 Kenyan population and housing census as baseline data. Aggregate population counts for census areas are re-distributed within each boundary onto 100 by 100m grid cells.

As shown in Fig 1 we followed an iterative approach and validated and refined the distribution from our geospatial analysis with results from relevant studies [46–48], sector publications [49] and information provided by sanitation stakeholders in Nairobi [50].

### Climate change impacts on current sanitation service configuration in Nairobi

As a next step, we combined descriptive climate scenarios for East Africa [51,52] and results of a recent systematic review on climate impacts on urban sanitation systems [2] to generally describe the likely impacts of climate change on the different sanitation service regimes in Nairobi. Acknowledging the uncertainty surrounding Kenya's climate trajectory in the coming decades, we considered the effect of different climate scenarios. Under the Future Climate for Africa research, the HyCRISTAL project delineated three basic scenarios of how climate change could be felt in urban East Africa by 2050 (Table 1) [52]: i) *Much wetter, large increase in heavy rainfall and hotter*; ii) *Increase in extreme rainfall and hotter;* iii) *Much hotter and drier with more erratic rainy seasons*.

We, combined the HyCRISTAL with insights on climate change impacts on urban sanitation failures from a recent systematic review [2] to contextualise the effect climate change is likely to have on Nairobi's stressed sanitation infrastructure across all sanitation service regimes.

### Analysis of policy and planning framework

To gauge the current trajectory of climate adaptation and sanitation, we conducted a comprehensive document review of relevant (national and county-level) laws, policies, strategies, and planning documents that are likely to shape future

**Table 1. Summary of 2050 climate scenarios for East Africa according to HyCRISTAL research.**

| | Future 1: Much wetter, large increase in heavy rainfall and hotter | Future 2: Increase in extreme rainfall and hotter | Future 3: Much hotter and drier with more erratic rainy seasons |
|---|---|---|---|
| Description | Increased rainfall during both long and short rainy seasons; More intense rainfall and increased frequency of storms; Average annual temperature increases of about 2°C and hotter temperature extremes. | No substantial changes in total rainfall amounts but more intense rainfall events and more frequent occurrence of extreme storms, longer dry spells, Average annual temperature increase of about 2–3°C and hotter temperature extremes. | Substantial increase of average annual temperatures (by about 3°C) and more frequent occurrence of extreme heat. Decrease in total annual rainfall amount and increase duration of dry spells and more frequent occurrence of drought events; Occasional occurrence of extreme rainfall and storms |
| Most critical climate change effect on the sanitation system | More frequent occurrence of recurrent and rapid onset floods (flash floods) | More frequent occurrence of flash floods | Water shortages; Occasional occurrence of flash floods |

Source: Summary based on [51].

sanitation (adaptation) planning in Nairobi. Apart from sanitation (or water sector) specific documents, we also included general framework documents such as the Constitution of Kenya and the Vision 2030 national development blueprint, as well as documents guiding the country's and county's approach to climate adaptation (e.g., the Climate Change Act and the Nairobi City County Climate Change Action Plan). We also reviewed available information on current and planned investments into Nairobi's sanitation sector, considering the strategic planning and budget documents of Nairobi City County (NCC), Nairobi City Water and Sewerage Company (NCWSC) and Athi Water Works Development Agency (AWWDA) and ongoing and planned investment of development partners (many of which are captured in NCC, NCWSC or AWWDA planning). We used the SEI Aid Atlas [53] as a starting point to identify relevant ongoing or planned investments made by various donors to Kenya. We subsequently examined detailed investment information on the websites of the specific donors and relevant investment databases African Water Facility (AWF), African Development Bank (AfDB) and World Bank (WB). We validated our selection with information from the Nairobi Integrated Urban Development Master Plan [54] and NCWSC, Water Sector Trust Fund (WSTF) representatives and non-public actors such as Sanergy and WSUP.

As no coherent definition for 'just' or 'equitable' climate adaptation of urban sanitation currently exists, we drew upon environmental and energy justice scholarship [22]. We adapted the energy justice framework proposed by Wood and Roelich [22] to the urban sanitation context by translating its justice dimensions into evaluative aspects of equitable sanitation adaptation (Table 2). We applied these evaluation criteria for deductive coding in NVivo across all information on sanitation planning, investments and climate adaptation.

## Estimate of Nairobi's sanitation future for 2030

Finally, we discuss plausible sanitation equity outcomes for sanitation provision in Nairobi 2030. We refer to our analysis of available policy, planning and investment documents to construct a sanitation service regime configuration for 2030

**Table 2. Aspects of equitable sanitation adaptation (based on Wood and Roelich [22]).**

| Tenets | Evaluation criteria relevant for equitable sanitation adaptation |
|---|---|
| **Distribution** *How are the benefits and burdens of sanitation provision (or failures) distributed through space and time?* | Which sanitation service configurations are represented in the current or planned adaptation measures? Who bears the environmental/public health risk before the investment/in case of failure of the measure? Does the adaptation measure change the distribution of environmental/public health risks between the different service regimes? Is there explicit recognition of aspects of intergeneration justice concerning environmental and public health risks? Who bears the financial risk of the current or planned adaptation investment? Is there explicit recognition of aspects of intergeneration justice concerning the financing mechanism? |
| *Recognition* *How are population groups and individuals represented and recognised in reference to sanitation adaptation* | Do the urban sanitation (adaptation) initiatives acknowledge and prioritise the needs of marginalised and vulnerable population groups? Do legal frameworks recognise and protect the rights of individuals and marginalised population groups to sanitation services, holding governments and institutions accountable? Is there evidence whether the current or planned measure explicitly considers affordability for different population groups? |
| *Procedures* *How are the processes involved in sanitation adaptation ameliorating injustices that have been identified* | Is there any evidence that the planning and decision-making process of the adaptation of the sanitation system considers structural inequalities in participation? Do the planning and decision-making process include mechanisms for meaningful engagement with local populations, ensuring their needs and concerns are considered in sanitation adaptation planning? Are reasons for inclusion/exclusion/prioritisation of adaptation measures concerning specific service configuration clear? |

and estimate the sanitation equity implications, considering the likely impacts of climate change and planned adaptive measures to counter these. We use rasterised population data from WorldPop (2000–2022) to extrapolate the spatial distribution of Nairobi's population in 2030 [55]. Acknowledging the different growth patterns in distinct parts of Nairobi, we used the ArcGIS Pro time series forecasting tools that enabled us to evaluate different extrapolation methods against each other and choose the best-fit population trend for each sub-area by validation or overall fitness of observed values [56]. We provide a detailed description of the forecasting method in the supplemental information (S4 File).

## Results and discussion

### Nairobi's splintered sanitation sector regime

Nairobi, a rapidly expanding city, grapples with significant urban disparities [40]. The historical context of its urban water sector, marked by deliberate, uneven development during colonial times [35,57] has left a complex legacy for post-independence water and sanitation planning. The city lacks an integrated system for managing and delivering sanitation services to the population. Instead, multiple actors and stakeholders provide sanitation services, often operating in isolation and frequently in some form of informality [36].

We use the conceptualisation of the sanitation service regime to unravel the unequal service distribution across Nairobi. We expand on Van Welie et al.'s [31] household-centric classification by incorporating neighbourhood outcomes (efficacy in excreta removal). Public and environmental health implications of sanitation occur in the household, neighbourhood, and broader cityscape [58]. We argue for a comprehensive analysis of distributive equity in sanitation services, considering both individual service realities and broader public health implications, at least on a neighbourhood scale. We acknowledge falling short of covering city-scale sanitation outcomes, as we do not assess the effectiveness and adequacy of treatment and disposal. Importantly, our classification of 'good sanitation' (as represented in the matrix bottom right) is not to be equated with safely managed sanitation in terms of the SDG target 6.2 (Fig 2).

According to a detailed urban excreta flow assessment, only 34% of Nairobi's excreta is safely managed [46,59], indicating substantial systemic failures across the sanitation service chain of all service regimes.

Residents lacking access to personal or communal toilets often resort to buckets, bags, or open defecation with (NT/NC or COPING regime) [31], with public toilets (PUBLIC TOILET/ PT regime) offering limited improvement from a public health perspective but failing to provide adequate individual sanitation. Furthermore, reliance on public toilets often coexists with other coping mechanisms, especially at night. In Nairobi, both options are associated with low-income, high-density slum settings.

Notably, many low-income areas, such as parts of Kibera or Mathare, are within NCWSC's service area but remain disconnected *ostensibly* due to costs, legal and infrastructural barriers, and inconsistent water supply [40,60].

Informal service providers organised in 'cartels' often control the water supply, sanitation emptying and solid waste collection in many informal settlements. The cartels control the pricing of basic services, restrict alternative service providers (including NCWSC staff) in their operations (through illegal charges and violence), intimidate residents, and are linked to deliberate acts of vandalism of sanitation infrastructure that threatens their business model [61–64]. Most public toilets in low-income areas, often managed by community organisations, are connected to the sewer system but are poorly maintained, failing to provide adequate services [65].

Shared toilets (ST) on plots inhabited by tenants are probably the most common sanitation solution for low-income residents in Nairobi. These toilets are either connected to the sewer system (ST/SEWER) or have some form of underground storage (pit/tanks). Due to the rapid densification of the urban areas in Nairobi, most of the non-sewered facilities get emptied once full, mostly resulting in indiscriminate and unsafe disposal of faecal sludge (ST/NC regime), often in proximity to the point of emptying [65]. Shared sanitation is typically associated with high-density single-storey 'shacks' in low-income settlements, but as Mwau and Sverdlik [60] point out, as Nairobi densifies, low-income accommodation is becoming increasingly heterogeneous and there has been a rise in high-rise (walk-up) low-quality tenement buildings

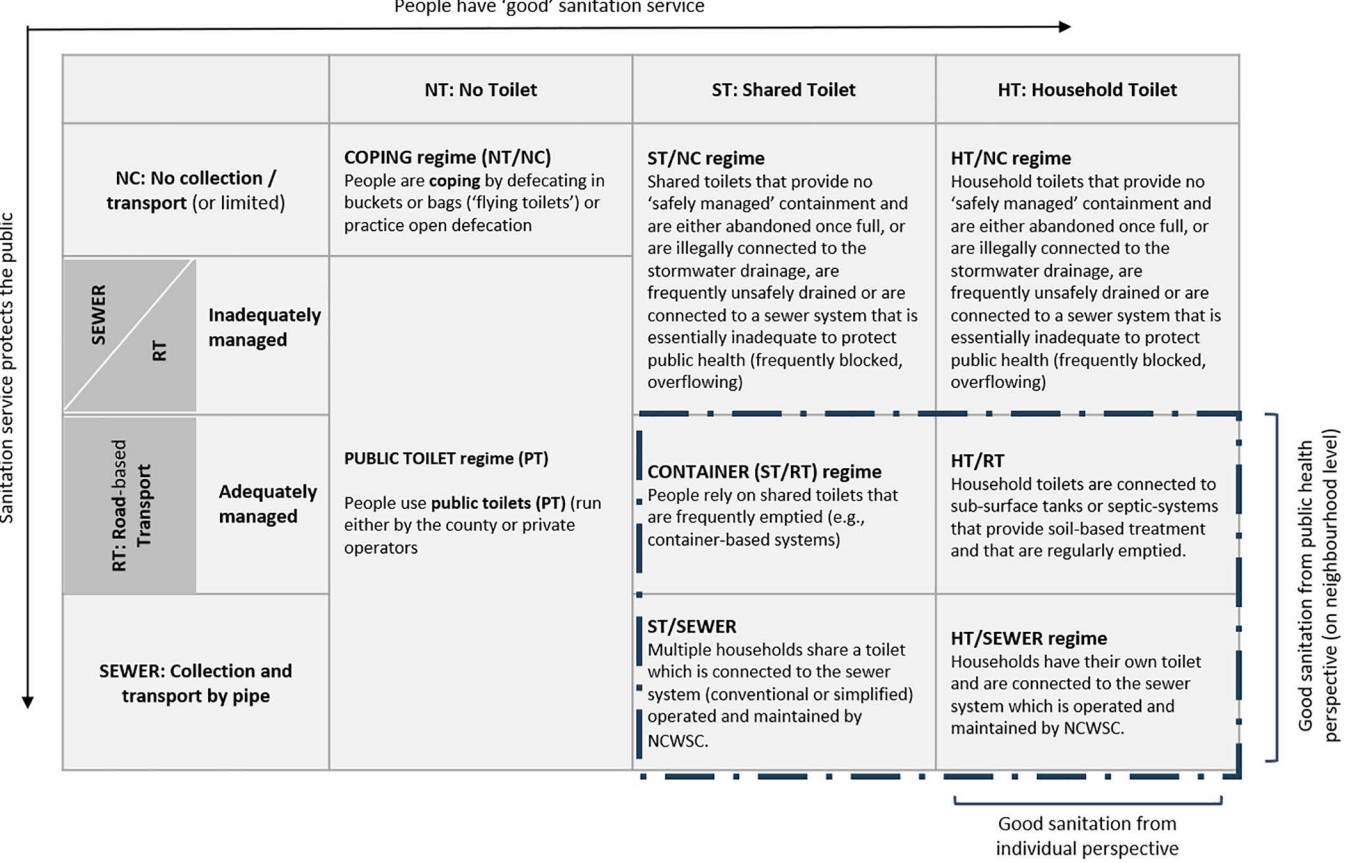

**Fig 2. Matrix showing distributional equity implications of sanitation service regimes using Nairobi as an example.**

with single-room rented accommodation and shared sanitation facilities. High-density tenement developments have put additional strain on the existing water and sewer systems and are frequently connected to informal sewers that drain raw sewerage into rivers such as the Mathare River [60].

Regular and formalised emptying for shared toilets is primarily provided for container-based sanitation systems run under the Fresh Life (FL) franchise, the non-profit entity of the international social enterprise Sanergy. Most FL toilets are shared by tenants of low-income housing plots and operated by the landlord [33]. Toilets shared between more than two households are not counted as safely managed under the JMP SDG monitoring. However, scholars and practitioners have long highlighted the need to recognise well-managed shared toilets as an important intermediate step for reducing sanitation inequalities in dense urban settlements, albeit also acknowledging concerns about the lack of nuance in knowledge about the realities of 'shared access' [66–68]. For simplification, we assume that currently, all toilets within the frequently emptied shared toilets regime in Nairobi are container-based systems operated by FL (CONTAINER).

We want to stress that inadequately managed road-based and poorly maintained sewer systems can pose a direct public health risk to urban residents. Nairobi's sewer system is partly dilapidated and lacks maintenance [63,69]. Sewer blockages, seepage and overflows distribute excreta within urban neighbourhoods with potentially similar adverse public health outcomes as overflowing pit latrines. In Nairobi, sewer seepage is mainly reported as a health risk in low-income areas [57]. Still, there is relatively little data on the citywide distribution and extent of the problem.

The Nairobi City Water and Sewerage Company (NCWSC) provides water and sewerage services in Nairobi. Generally, household toilets connected to the sewer system (HT/SEWER) are associated with high-income or middle-income neighbourhoods (often apartment blocks) in the centre of the town. Yet, the city's sewer system, outdated and overburdened, struggles to keep pace with urban growth. The Dandora and Kariobangi wastewater treatment plants, operated by NCWSC, suffer from poor maintenance, leading to inefficient wastewater treatment and environmental pollution [70,71].

Contrary to common belief, not all middle and high-income residences in Nairobi are connected to the sewer system. Many low-density affluent neighbourhoods, such as Karen or Muthaiga, are located outside the sewered area and rely on household toilets connected to underground tank systems (HT/RT). These toilets provide a good service for the individual and, due to the low density of the areas, are unlikely to pose a direct public health threat in the vicinity of use. However, we want to stress that they often do not meet the technical criteria of septic systems and are often served by informal exhauster services, largely controlled by cartels who frequently illegally dump their load into the environment [57,61,63]. As such, these toilets might still be counted as not being safely managed according to the JMP criteria [72].

## Spatial and quantitative distribution of sanitation services in Nairobi

Following the allocation process described in Fig 1 we arrived at an approximate current spatial distribution of sanitation service regimes, underscoring the complexity of addressing urban sanitation (Fig 3).

Using population raster data, we estimated the approximate quantitative population distribution per service regime and compared our results against available sanitation data. We validated our data against multiple available sources and found that our estimates (Table 3) are roughly in line with data on safely and unsafely managed sanitation in Nairobi [46], sewer coverage [73] as well as emptying practices of onsite sanitation and use of shared sanitation facilities [74].

Our estimates show that less than a third of the population in Nairobi is provided with a 'good' sanitation service from an individual and neighbourhood perspective (as constituted by the HT/RT or HT/SEWER regimes). We estimate that around 40% of the population relies on services that are problematic from the perspective of personal safety and convenience or public health protection in urban neighbourhoods (COPING, PUBLIC TOILET, ST/NC or HT/NC). We would also like to highlight that whilst we treat shared sanitation service regimes that effectively remove excreta from the neighbourhood (CONTAINER and ST/SEWER) as acceptable intermediate solutions, these systems do not always constitute good sanitation from an individual and public health perspective. We have already discussed the public health risks from sewage seepage or overflow. In addition, particularly for low-income and informal areas, there are reports of exceedingly high numbers of people sharing toilets [47,75] and the regulator noted a trend for a rising number of people per sewer connection which raises concern in terms of the quality for the user [73]. Getting reliable estimates of the number of container-based sanitation units is challenging. Still, in October 2023, a Fig of 5,500 was provided, although, by Fresh Life's own admission, the numbers fluctuate monthly. Emptying of these units is done under the control of Sanergy, and hence, public health risks are minimised. Most units are shared between multiple households. The Sanergy system continues to be substantially subsidised by donor funding, and so far, attempts to agree with NCC on a publicly subsidised service contract have not been successful [33,36,65].

## Climate change effects on failures and distributional inequity within Nairobi sanitation system

Having outlined the current sanitation situation in Nairobi and some of the general challenges and failures of the current sectoral regime, we will now analyse how climate change will likely affect the sanitation service situation in Nairobi.

Table 4 summarises relevant climate impacts along the sanitation service chain for each service regime, highlighting existing pressure.

The synthesis shows that for all sanitation service regimes in Nairobi, flooding – especially flash floods – emerges as a critical concern under all three climate scenarios. In Nairobi, stormwater is managed through both natural and artificial

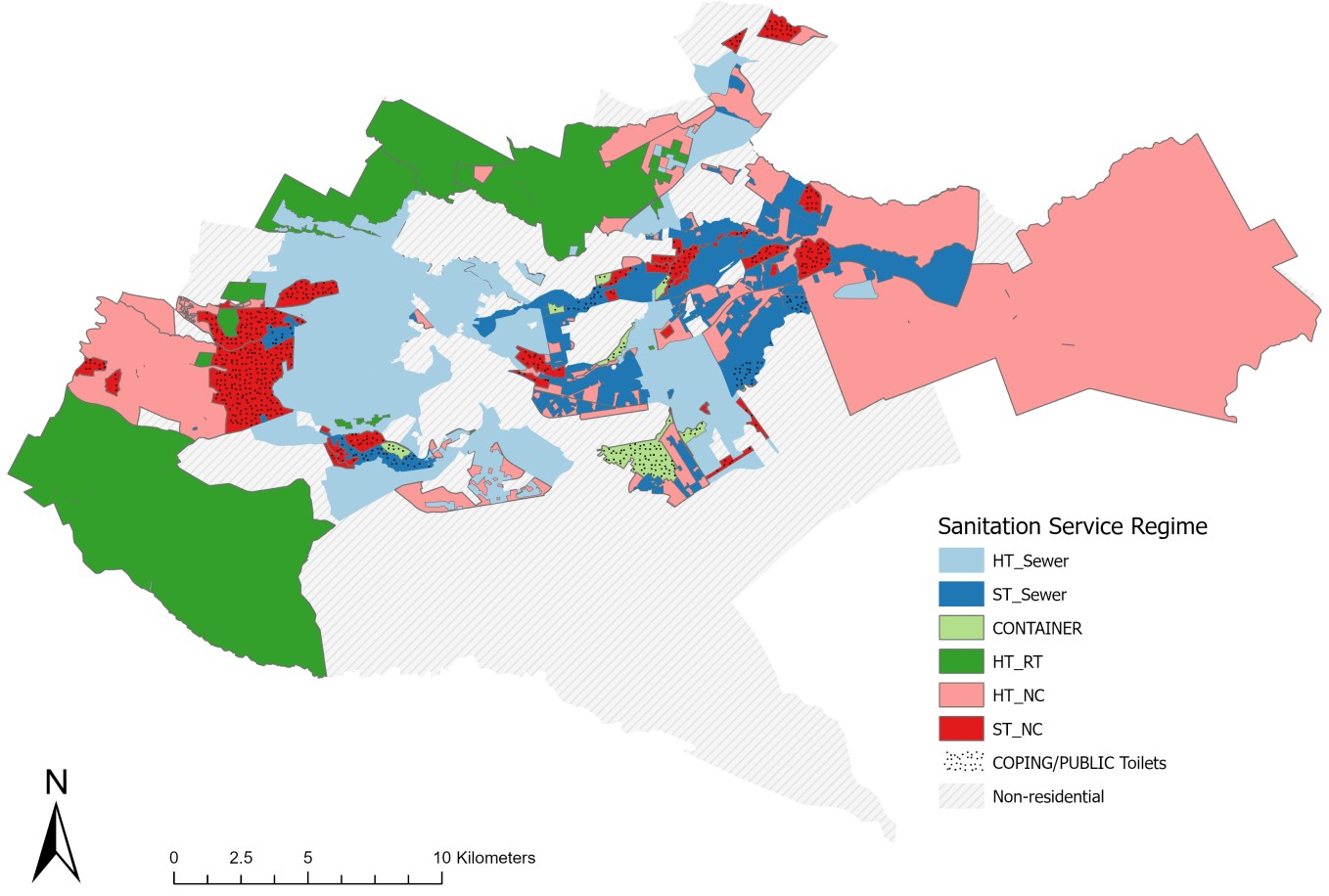

**Fig 3. Approximate spatial distribution of dominant sanitation service regimes in Nairobi – present situation: COPING (NC/NT) = People don't have access to toilets us coping mechanisms; PUBLIC TOILET = People rely on public toilets as main form of sanitation; ST/NC = Shared toilets that are NOT adequately managed; CONTAINER (ST/RT) = Shared toilets that are regularly emptied; ST/SEWER = Shared toilets connected to the sewer system; HT/NC = Household that are not adequately managed; HT/RT = Household toilets that regularly emptied; HT/SEWER = Household toilets connected to the sewer system.**

**Table 3. Approximate quantitative distribution of population per sanitation service regime in Nairobi – present situation rounded to 10,000s (based on 2022 population raster data [45] and sanitation service regime allocation given in Fig 3).**

|  | COPING | PUBLIC TOILET | ST/NC | CONTAINER | ST/SEWER | HT/NC | HT/RT | HT/SEWER | SUM |
|---|---|---|---|---|---|---|---|---|---|
| % | 2 | 2 | 10 | 3 | 23 | 30 | 8 | 23 | 100 |
| Total | 70,000 | 70,000 | 450,000 | 150,000 | 1,050,000 | 1,370,000 | 350,000 | 1,050,000 | 4,560,000 |

drainage systems. Artificial stormwater drainage includes (where available) combined sewer pipes and roadside drains. Stormwater drains are frequently blocked by solid waste, leaves and debris, and the lack of maintenance of Nairobi's drainage system is a longstanding problem [44,76]. With substantial technical data on the existing stormwater drainage system being lost, there is no comprehensive urban stormwater management plan so far [77,78]. In addition, some pit latrines – or other toilets relying on subsurface containment – are illegally drained during heavy rainfalls and floods, contributing to neighbourhood-wide faecal contamination. Exhauster trucks and other emptying services get disrupted when

**Table 4. Relevance of climate change impacts along the sanitation chain (based on [2] under different future climate scenarios established by the HyCRISTAL.**

| Service Regime | Access/ Containment | Emptying/ Transport | Treatment | Disposal/ Enduse |
|---|---|---|---|---|
| HT/SEWER | Alternative (potentially unsafe) sanitation options are used as coping mechanisms due to failing water-based toilets during periods of water restrictions **F3** | Increased risk of pipe damage due to changed soil moisture and subsidence **F1** Increased inflow and infiltration into the sewer system (particularly into dilapidated pipes and through missing manhole covers) | Flooding and damage to WWTPs structure and equipment Flooding of WWTP leading to temporary system failure and discharge of raw sewage Electricity failure leading to failure of pumps and aeration | Discharge of untreated/partially treated effluent due to overloading or bypassing of treatment Contamination of receiving water bodies due to WWTP failure **F1**, F2, F3 |
| ST/SEWER | Damage to superstructure making toilet unusable Toilets become inundated/ inaccessible (causing people to (temporarily) abandon toilet **F1**, F2 Alternative (potentially unsafe) sanitation options are used as coping mechanisms due to failing water-based toilets during periods of water restrictions **F3** | Sewer blockages after and event because of sand, debris or solid waste entering sewers and pump stations Electricity failure leading to failure of pumps Damage to sewer mains and pumps Overload of sewer system resulting in overflow to the drainage system **F1**, F2, F3 Higher risk of blockages in the sewer and drainage system and discharge pipes Higher risk of corrosion of sewers Pipe and joint breakages through ground settlement after prolonged droughts (+) Decreased risk of urban flooding **F3** | Road interruptions leading to disruption of site access for WWTP staff and supplies Pollutant load exceeding biological treatment capacity of WWTP Reduced nutrient removal capacity during high intensity rainfall **F1**, F2, F3 Groundwater inflow and infiltration into wastewater pipes causes higher inflow into WWTPs and further stretches their insufficient treatment capacity **F1** Higher concentration of wastewater leading to less effective treatment Corrosive influent damages equipment in treatment plants Excess deposition due to low flow **F3** Moderate increases in temperature might increase efficiency of biological WWT F1, F2, F3 | (+) Increased dilution of influent **F1** Less dilution in receiving waters **F3** |
| HT/RT | Backflow of sewage from septic tanks Damage to pits, septic tanks and absorption fields Inundation of drain fields **F1**, F2, F3 Floatation and damage of septic tanks due to high groundwater table **F1** Low moisture content of soil leading to erosion and damage of subsurface structures **F3** | People 'drain' toilets during flood events leading to faecal contamination Limited access for emptying services due to: - Structural damage to pavements and other road elements such as bridges - Road collapse or development of sinkholes due to destabilisation of soil caused by damaged sewer pipes Roads become inundated/inaccessible - Decreased road capacity/ increased congestion/ increased travel time **F1**, F2, F3 Limited access for emptying services due to heat damage to access roads F1, F2, **F3** | Flooding and damage to FSTP Road interruptions leading to disruption of site access for FSTP staff and supplies Flooding and damage to wetland flora **F1**, F2, F3 (+) Moderate increases in temperature might increase efficiency of FST in septic tank systems and FSTPs F1, F2, F3 | Higher groundwater/ surface water pollutant risk caused by increased mobility of pollutants from septic tank drainfields **F1** |

*(Continued)*

**Table 4.** (Continued)

| Service Regime | Access/ Containment | Emptying/ Transport | Treatment | Disposal/ Enduse |
|---|---|---|---|---|
| ST/NC and HT/NC | *For septic systems see above* Damage to superstructure making toilet unusable Toilets become inundated/inaccessible (causing people to (temporarily) abandon toilet Damage/collapse of pits Flooding of pits from below **F1,** F2, F3 Low moisture content of soil leading to erosion and damage of subsurface structures (+) Lower GW pollution risk from pit latrines **F3** | People 'drain' toilets during flood events leading to faecal contamination **F1,** F2, F3 | (+) Moderate increases in temperature might increase efficiency of FST in septic tank systems and FSTPs F1, F2, F3 | Higher groundwater/surface water pollutant risk caused by increased mobility of pollutants from septic system drainfields **F1** |
| CONTAINER (ST/RT) | Damage to superstructure making toilet unusable Toilets become inundated/inaccessible (causing people to (temporarily) abandon toilet **F1,** F2, F3 | Limited access for emptying services due to: - Structural damage to pavements and other road elements such as bridges - Road collapse or development of sinkholes due to destabilisation of soil caused by damaged sewer pipes Roads become inundated/inaccessible - Decreased road capacity/ increased congestion/ increased travel time **F1,** F2, F3 Limited access for emptying services due to heat damage to access roads F1, F2, **F3** | Flooding and damage to treatment facility Road interruptions leading to disruption of site access for treatment facility staff and supplies Flooding and damage to wetland flora **F1,** F2, F3 | |
| PUBLIC TOILET (PT) | Damage to superstructure making toilet unusable Toilets or access roads become inundated/inaccessible (causing people to (temporarily) abandon toilet **F1,** F2, F3 | | *Depending if connected to sewer or septic system – refer to sections above* | |
| COPING (NT/NC) | General unhygienic conditions during flooding increase risk of faecal contamination of surroundings from open defecation and other unsafe sanitation practices **F1,** F2, F3 | | *N/A* | |

Likely to be relevant for specific climate future scenario: F1 - Future 1: Much wetter, large increase in extreme rainfall and hotter; F2 - Future 2: Increase in extreme rainfall and hotter; F3 - Future 3: Much hotter and drier with more erratic rainy seasons (**bold:** likely to be highly releveant for specific climate future scenario)

(+) Positive climate impact

roads are flooded or damaged during extreme rainfall [61]. People who do not have toilets within their accommodation might have temporarily limited or no access to their toilets during floods or lose access completely if toilets get damaged. Untreated or partially treated sewage (from sewer overflows or overloaded treatment facilities) contributes to environmental and particularly river pollution [79].

Urban flood modelling is complex and often restricted by the coarse granularity of global models, which neither sufficiently capture susceptibility factors nor are these models suitable to assess real vulnerability or adaptive capacity of the exposed urban areas [80–82]. We compared the 2010 GIS land use classifications by Ledant [32] with a current basemap showing further urban densification along flood- and landslide-prone riverbanks. Major housing developments (formal

and informal) have occurred in many formerly undeveloped or agricultural areas in Nairobi. It is expected that most of the population increase will be in the low and middle-income areas in the eastern parts of the city [40,83], such as the Kasarani sub-county which are highly susceptible to frequent flooding even under current climatic conditions and where rapid property development will also substantially increase the proportion of paved surface area [84].

More erratic rainfall patterns leading to prolonged duration of dry periods or drought will particularly stress the effective operation of piped-based systems (HT/SEWER and ST/SEWER) as they are likely to contribute to higher risk of blockages and corrosion in sewers and treatment facilities and lower dilution capacity of wastewater receiving waters amongst others [2]. NCWSC is currently unable to meet the population's water demand [85]. As demand has outstripped water supply in Nairobi for decades, Ledant [32] attests that "Nairobi is in a situation of structural water shortage likely to worsen until further expansion" (32, p.338). While substantial investments have been made into Nairobi's water supply system [85], climate change is already an additional stressor for the water supply and sanitation systems. In the last decade, Kenya has suffered from extensive drought, which has affected Nairobi's water supply. The supply rationing led to a drop in NCWSC's revenue, which contributed to reduced budget allocation for sewer expansions in the informal settlements [85]. Whilst middle and higher-income customers often have sufficient domestic water storage tanks to cope with interrupted water supply, low-income residents often do not have the same backup and therefore, sewer systems in low-income areas (ST/SEWER) will be most severely affected by increased water shortage under climate change [86]. Simplified sewer systems (SSSs) that require less water and are cheaper to implement have been piloted in parts of Mukuru informal settlements. SSSs are not a recent technology and are widely used globally, mainly in Latin America. The existence of trunk sewers in many low-income areas of Nairobi offers favourable implementation opportunities. However, a robust evaluation of the scalability and sustainability of these systems in Nairobi, particularly concerning institutional willingness and financial allocations, needs further research [87].

## Gaps between policy and implementation

In the following sections, we present the findings of our review of legal, policy, and investment documents, focusing on how these documents recognise and address equity concerns. However, we would like to embed our analysis in a more general debate of gaps between policy and implementation – which is not a Kenya or sanitation-specific problem but a global phenomenon observable at all levels of state or organisational governance.

In urban planning and, specifically, climate resilience, the key to understanding these gaps is the concept of 'fantasy plans' [88], describing policy aspirations that are ambitious and possibly idealistic but disconnected from practical and achievable implementation strategies. This often leads to formulating policies that are more symbolic and performative than substantive, creating an illusion of progress without tangible results [88,89]. In the context of basic service delivery, on paper, such policies appear to recognise equity challenges; however, by adopting policies that are not grounded in reality, governments risk perpetuating inequities and failing to change the distributional pattern of sanitation services genuinely. In the context of climate adaptation, such 'fantasy plans' may be further perpetuated by governments' efforts to align with and conform to donor-driven adaptation and mitigation goals, which risk greenwashing business-as-usual approaches [17].

## Linking climate change and sanitation in policy and planning –mainstreaming or out-of-sight?

Our review showed that within the current national and county-specific legal and policy framework, climate adaptation is not adequately reflected – and budgeted for – in sanitation plans, and sanitation is barely considered in climate change adaptation law and policies.

The National Climate Change Act [90] requires that County Governments mainstream climate change actions and interventions in their County Integrated Development Plans (CIDPs) but does not reference the water and sanitation sector. NCC's current CIDP makes no concrete links between climate adaptation and sanitation. Hence, no climate resilience,

adaptation, or mitigation strategies exist for the sanitation sector within the current county planning [91]. On the national level, there are no specific sanitation-related commitments to adaptation or mitigation in Kenya's Nationally Determined Contributions (NDCs) [92]. The National Adaptation Plan (NAP) [93] offers only vague actions for the water and sanitation sector (S1 Table). It links long-term actions to the National Water Master Plan [94], which focuses on water resource management. As such, it lacks a specific focus on climate adaptation within the sanitation sector in the Athi Catchment.

Within the water and sanitation sector, specific legal and policy framework (S2 Table) climate change adaptation is primarily considered (if at all) in the context of water resources/water security and (less frequently) urban drainage or flood resilience. Two of the most recent sector documents – the National Water and Sanitation Service Strategy (2020– 2025) [95] and the National Sanitation Management Policy (the reviewed document is a report documenting the process of developing the policy available as the draft policy that is currently debated could not be obtained) [96] – show an increased recognition of the potential impacts of climate change on the sanitation sector.

The supplemental information (S1-S3 Table) lists all reviewed legal, policy and planning documents and summarises their respective consideration of climate change adaptation in the context of urban sanitation.

### Is there any money for climate adaptation of sanitation?

Evaluating budget allocations offers insight into governments' commitment to equitable and long-term provision of basic services beyond performative pledges. A detailed assessment of the climate finance landscape in Kenya [97] revealed a primary government focus on investments in mitigation efforts, especially in the renewable energy sector. This is despite the NAP [93] and the NDCs [92] emphasising adaptation strategies. Although investment in the water and wastewater management sector was the most substantial in terms of adaptation in 2017/18, it still falls short of the requirements for water sector adaptation outlined in the NAP [93]. Importantly, the climate finance landscape assessment has no further subdivision between the water and wastewater sub-sectors. Given the current policy focus (see above), it is probable that most investments were directed towards initiatives to alleviate water scarcity.

On the county level, the apparent disconnect between sanitation and climate change policy, as discussed earlier, aligns with the absence of explicit links between climate change and urban sanitation in Nairobi City County's (NCC) urban planning and budgeting frameworks [69,91,98]. Under the institutional framework of the Kenyan water sector, the Athi Water Works Development Agency (AWWDA) is responsible for developing sewerage infrastructure, whilst Nairobi City Water and Sewerage Company (NCWSC) manages and maintains water and sewerage services in Nairobi City County [99]. Both entities display only limited and general references to climate change adaptation in their most recent investment plans [85,100].

Kenya's water sector heavily depends on donors, with approximately 70% of the annual capital investment coming from development partners [101]. In addition, Kenya expects that implementing their NDCs will require international partners to sustain at least 87% of the costs by 2030, a currently unmet target [97].

The reviewed investments by the development partners (S3 Table) all incorporate considerations for climate resilience in infrastructure design, primarily aimed at mitigating flood risks through enhanced drainage and sewer systems. However, none of the projects explicitly mention climate change adaptation in their outcome indicators [70,102–105]. The Green Climate Fund's Infrastructure Climate Resilient Fund does not directly finance the sanitation sector. While its proposed investments in road and bridge rehabilitation could indirectly benefit sanitation services dependent on road transport, specific details about the locations of these investments remain unclear [106].

### *Adaptive sanitation services for whom?* Equity considerations in the current sanitation and climate change adaptation planning framework

We now turn towards a deeper examination of the equity considerations within the policy and planning framework for sanitation climate adaptation in Nairobi. Our initial spatial and quantitative analysis highlighted existing distributional inequities within Nairobi's present sanitation service regime configuration. As previously noted, climate change will put an uneven

burden on urban populations and is likely to worsen the challenges of low-income and marginalised groups and exacerbate existing inequities [107,108].

We have found a broad disjoint between climate adaptation and sanitation service planning in the current policy framework and budget allocations. Although climate change adaptation is not explicitly integrated into sanitation policies, and vice versa, addressing urban vulnerabilities in both areas could still help prevent climate change from intensifying sanitation inequities. Thus, in the following, we explore if and how the three dimensions of sanitation adaptation equity (Table 2) are incorporated within the policy, planning and investment framework.

Our findings suggest that, despite formal recognition of sanitation service inequalities and the needs of marginalised groups within various policies, this awareness seldom translates into actions prioritising and effectively enhancing distributive equity in sanitation service across the city. This echoes Weinstein et. al's [88] assessment of flood resilience planning in Mumbai: "Yet while the discourse has shifted to include some language [...], the specific developments it proposes do not suggest an altered development context" (88, p.274) Moreover, we recognise that assessing procedural equity is challenging when solely analysing legal and policy documents. We draw on existing scholarly work to explore the misalignments within Nairobi's divided sanitation system. This analysis highlights how these misalignments further entrench sanitation inequities within the context of broader structural and procedural inequities

**Recognition.** Over the last two decades, Kenya has made some progress in recognising the needs of low-income populations within its water and sanitation governance framework. The Water Sector Trust Fund was established in 2002 to increase investment in water and sanitation for low-income and marginalised population groups. The National Water Services Strategy (2007–2015) and the Pro-Poor Implementation Plan for Water Supply and Sanitation, adopted in 2007 [109], signalled a commitment to a pro-poor approach in water and sanitation service delivery. The Kenyan constitution, amended in 2010, includes the human right to water and sanitation [34,110]. In 2018, the national water sector regulator WASREB introduced a 'Pro-poor' Service Assessment in the annual utility reporting [111]. However, the respective indicator is not part of the core performance ranking metrics. Most sanitation investments in low-income areas heavily rely on external funding [34]. Acknowledging the persisting funding gap for sanitation development, there have been longstanding discussions in Kenya about introducing a levy to fund the sewer system expansion and investment in non-sewered sanitation [95,99]. In 2019, WASREB has published guidelines for structuring such a levy [112]. However, no Water Service Provider (WSP) has implemented this levy, and its execution appears unlikely due to the recent backlash against the housing levy and the ongoing global cost of living crisis [113].

Similarly, the climate change-related policy framework generally recognises the vulnerability of poor urban neighbourhoods (e.g., NCCRS, 2010 [114], NCCAP, 2018 [115]) but pledges to fair distribution, such as the NPCF's (2016) commitment to 'equitable benefits sharing' remain vague and unspecific.

**Distributional.** Within the sanitation policy and planning framework of Nairobi, the identified references to the concept of distributional equity are primarily related to the evaluation criterion: *'Which sanitation service configurations are represented in the current or planned adaptation* measures' (Table 2). Notably, a considerable portion of the legal and policy framework, including the National Water Master Plan (NWMP) [94], the Water Act [99], the CIDP [91] and the Nairobi Integrated Urban Development Plan (NIUPLAN) [69] predominantly overlooks non-sewered sanitation service regimes. This omission effectively excludes populations for whom a connection to the sewer system is not attainable in the near future

The prevailing sanitation sector investment strategy, adhering to the paradigm of the *'networked city'* [116], concentrates on sewer network expansion aiming to increase sewerage coverage in Nairobi from 50% to 80% by 2030 [91] and on rehabilitating the existing sewer network that has suffered from insufficient maintenance for decades. Despite the ongoing and planned investments, such a substantial increase in coverage within this period appears unrealistic. Whilst the Kenyan Government has revised its national sewerage coverage target from a fantastical 80% by 2030 to a more balanced (but likely still over-ambitious) goal of 40% sewerage and 60% safe onsite sanitation [73], the NCC and NCWSC

have yet to adjust their targets. Due to rapid population growth, the development of sewerage infrastructure has not kept pace with this rapid population increase and therefore, the sewerage coverage rate has remained stagnant at 50% since 2015 [49,111]. Importantly, coverage also only describes the existence of a sewer network and should not be equated with people being connected to (i.e., accessing) sewered-based sanitation [117].

Plans and investments dedicated to improving the sanitation situation for those who will not be connected to the sewer network in the near future, especially in terms of climate resilience, are markedly limited. The water service provider and the county have shown limited consideration for heterogeneous sanitation service regime configuration. The NWMP [94] typifies the endemic vagueness characterising non-sewered sanitation planning in Nairobi. The NWMP's sanitation annex for the Athi catchment area, which includes Nairobi, vaguely declares:

> "Outside the sewerage service area, the improved on-site treatment facilities will be available for the remaining 4.28 million residents in 2030. […] unimproved facilities will be improved with new housing. Development of on-site sanitation facilities is planned for the ten counties in ACA." [94]

From the information in the NWMP however, it is not clear who will be responsible for planning, implementing, and paying for these improved on-site sanitation facilities. Although more recently, Kenya has (at least on paper) committed to Countywide Inclusive Sanitation (COWIS) planning [112,118], and a feasibility study for the Nairobi Inclusive Sanitation Improvement Project [105] is underway, a concrete vision for managing and funding non-sewered sanitation programs is still absent. Almost all public investments in non-sewered sanitation in Kenya depend on external funding both for capital and operational expenditures [33,112], illustrating their similarity to sewered sanitation systems, which rely on continuous public or other funding for sustained operations and maintenance. Thus, without government commitments to take on their operations, such systems remain unsustainable [9,119]

Unlike some other Kenyan water utilities – such as the utilities in Kisumu, Nakuru or Malindi – NCWSC has shown little enthusiasm for engaging in non-sewered sanitation service provision [33]. In their water and sanitation planning, NCC merely incorporates the sewerage-focused plans from the NCWSC and commits to constructing some public toilets [91], without showing any meaningful engagement with any non-sewered sanitation service regimes. Service providers such as NCWSC are often reluctant to engage in non-sewered sanitation or to work in low-income areas, often citing the lack of profitability of such endeavours, despite the lack of evidence that any sanitation interventions, including sewerage, can or indeed should be fully self-financing [119]. Various scholars have noted that the prevailing institutional incentives and regulatory frameworks prioritise utilities' operational and financial efficiency over the constitutional right to water and sanitation [32,120]. Kenya's urban water sector is undergoing a commercialisation process, and – despite rhetorical commitment to COWIS – service providers are encouraged to attract private sector investment and formulate 'bankable' projects not only by government policy but also by development partners [120,121]. In the absence of structured public finance for adequate sanitation service provision outside the sewered sanitation service regimes, the strong focus on cost-recovery conflicts with the public health benefits of adequate sanitation and has been identified as a major hindrance in promoting citywide sanitation [9].

**Procedural.** Regarding procedural equity, our analysis exposed predominately ambiguous rhetoric about 'inclusive planning' in policy formulation processes [e.g., 122] or equally unspecific commitments to 'extensive consultations' that should occur in further planning [e.g., 92]. The Nairobi City County (NCC) admits to the low public inclusivity in the previous County Integrated Development Plan (CIDP) (2018–2022) and broader NCC activities yet fails to outline specific strategies for enhancing public engagement in the implementation of the current CIDP [91]. Scholars have characterised Kenya's urban planning governance as 'exclusionary' and 'technocratic' [123,124], noting that Nairobi's urban development, including water and sanitation provision, remains influenced by colonial legacies of segregation and exclusion [35,124]. Despite growing recognition of non-sewered sanitation (see above), there is still no clear institutional home, coherent governance framework and political will to deal with the planning and management of

non-sewered sanitation, particularly in informal settlements [34,36]. Additionally, strong vested interests persist in maintaining unregulated informal sanitation services controlled by well-connected and exploitative 'cartels' in the low-income areas of Nairobi [61,125]. Sanitation planning in Nairobi is not well linked to other relevant urban sectors such as housing and land-use planning, transport and energy planning [48]. Even the interdependencies between sanitation with the most closely related urban sector, water supply, are not adequately considered. Water-based sewerage networks are still the only official sanitation strategy of NCWSC [85]. However, water demand in Nairobi has long outstripped supply. As a result, water rationing has become part of the official supply strategy of NCWSC and insufficient water pressure is an increasing challenge for quickly developed new high-rises for the middle class [60,126].

On paper the Mukuru Special Planning Area (SPA) presents a notable exception. The informal settlement upgrading project initiated in 2017 officially embraces all of the three aspects of sanitation adaptation equity, particularly in its inclusive processes. The designation of Mukuru as an SPA led to the suspension of conventional planning regulations, acknowledging their inadequacy for the settlement's unique challenges and enabling more context-specific solutions [123]. While not presented as a best-practice example, the Mukuru SPA's efforts in facilitating resident co-design of climate change adaptation measures, including improvements in flood resilience, drainage, and sanitation systems [127], are commendable within an overall exclusionary urban governance framework [123]. The NCWSC piloted a limited number of simplified sewer connections in Mukuru, which are more suitable for pour-flush toilets, intermittent water supply, and drought resilience [87]. Separately, Fresh Life provides container-based sanitation systems in the area.

However, the costs of the connections to the simplified network have been criticised by residents and local leaders [64] and many Mukuru residents still face inadequate sanitation, with high numbers sharing toilets and people reverting to buckets or other forms of unsafe sanitation due to the long distances to toilets, particularly at night [127,128]. The implementation of the co-designed SPA proposals has been delayed and is still ongoing, precluding a comprehensive evaluation. Generally, the value of the co-design process has been acknowledged by various researchers [123,127,129], although NCC engagement in facilitating the participation of residents has been criticised as insufficient [129] and residents of Mukuru still feel that the government does not adequately engage with improving their sanitation situation, prioritising instead the more affluent areas in the city [64]. One of the authors who visited Mukuru SPA in February 2024 could not identify any evidence to suggest systemic or transformative engagement of NCC in Mukuru.

In summary, in Nairobi, the sanitation policy and planning framework mimic 'recognition' of sanitation inequities and climate threats whilst failing to translate their ambitious and performative statements into actions that would genuinely enhance the procedural or distributive equity within the 'splintered sanitation sector'. Additionally, national policies in Kenya extravert 'pro-poor' inclusion within the water and sanitation sector as well as climate change mainstreaming. This is emblematic of a broader trend where development partners use aid conditions to incentivise governments to reproduce 'best practice' reforms of public sectors – such as the water sector – fostering the establishment of ostensibly 'ideal' institutions equipped with symbolic 'good' policy and institutional frameworks whilst lacking the requisite capability to fulfil their promises [130]. The performativity in Nairobi's water and sanitation sector and adaptation planning is not an isolated case of 'style over substance' incentivised by the international aid and financing system. These findings echo Narzetti's and Marques' [131] analysis of reforms of Brazilian water and sanitation who describe the ineffectiveness of the de jure well designed 'pro-poor' and 'inclusive' water sector policy framework that de facto fails the 100 million Brazilians without wastewater collection services and largely excludes the peri-urban population.

## Estimate of future conditions

The evidence from our projections of the future trajectory predominantly suggests a business-as-usual approach to sanitation service provision in Nairobi. This implies that substantial reconfiguration or realignment of sanitation service regimes is unlikely; therefore, the sector is expected to remain fragmented, with marked inequities in access to and quality of sanitation services.

 

Despite ongoing investments in conventional sewerage, which will increase the number of individuals with sewer access; primarily by connecting shared toilets to sewers (ST/SEWER), these efforts are projected to be substantially offset by rapid population growth in Nairobi. Assuming a 2030 population of 6 million [44], maintaining the current 50% sewerage coverage would require connecting approximately 700,000 people, or about 90,000 people annually from 2022 to 2030. To achieve the NCC target of 80% sewerage coverage by 2030, the number increases to 280,000 people annually.

Our analysis found no substantial evidence suggesting that non-sewered sanitation systems will be more effectively organised by 2030. Therefore, our projected spatial distribution for Nairobi in 2030 under a business-as-usual scenario incorporates the planned sewer extensions outlined in the Nairobi Integrated Urban Development Master Plan (NIUPLAN) [69]. We expect minimal changes to the overall spatial distribution of sanitation service regimes beyond these extensions (Fig 4).

The expected modest shifts in the spatial distribution of service regimes indicate that the proportional distribution of individuals using these services will largely depend on the geographic pattern of population growth. At the time of writing, the Kenya National Bureau of Statistics (KNBS) had not released official population projections based on the 2019 census

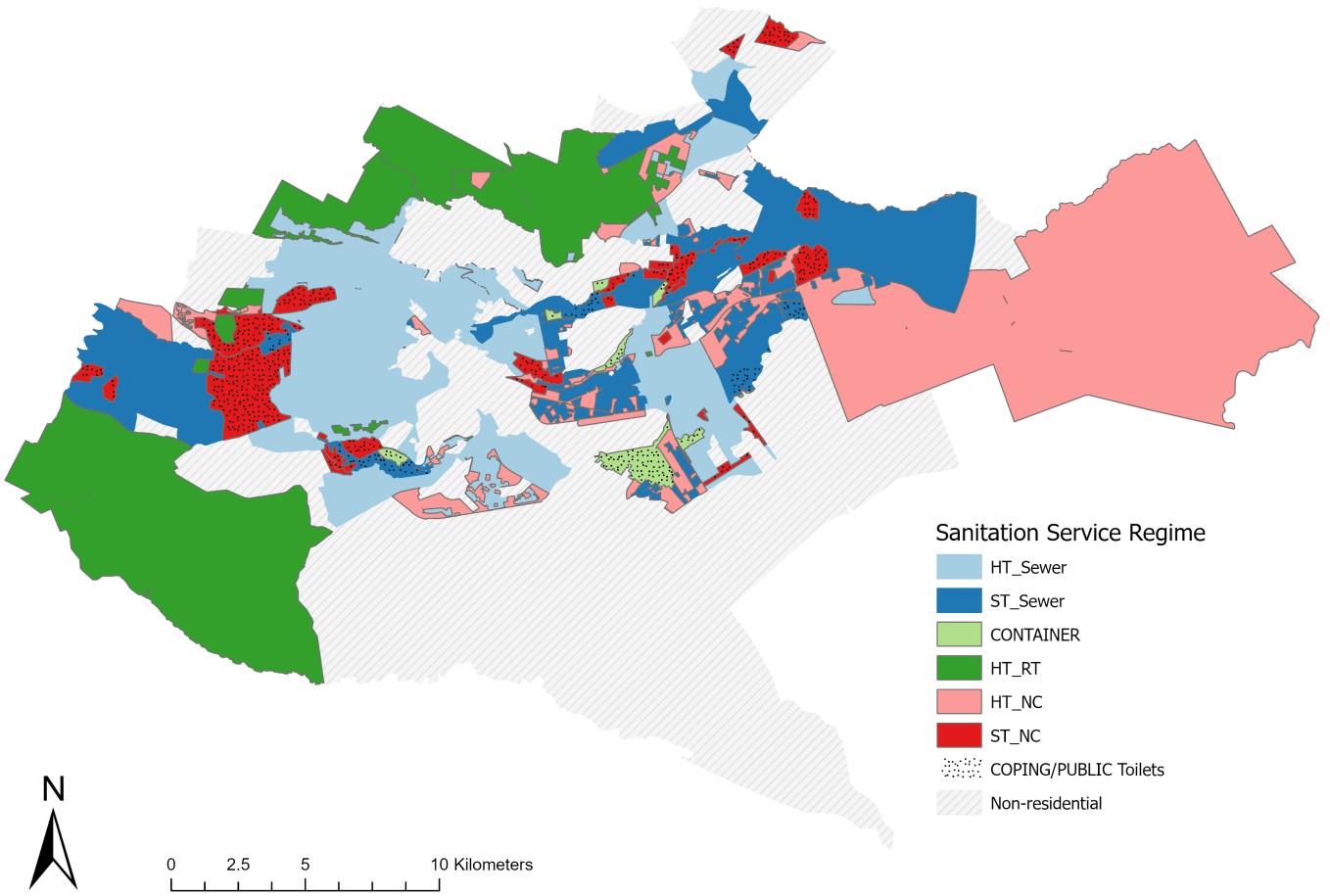

**Fig 4. Approximate spatial distribution of dominant sanitation service regimes in Nairobi – projection for 2030 (based on information about sewer development from Nairobi Master plan and planning framework detailed in S2 Table): COPING (NC/NT) = People don't have access to toilets us coping mechanisms; PUBLIC TOILET = People rely on public toilets as main form of sanitation; ST/NC = Shared toilets that are NOT adequately managed; CONTAINER (ST/RT) = Shared toilets that are regularly emptied; ST/SEWER = Shared toilets connected to the sewer system; HT/NC = Household that are not adequately managed; HT/RT = Household toilets that regularly emptied; HT/SEWER = Household toilets connected to the sewer system.**

data. Our total population Fig of 6.1 million is consistent with the estimate of 6 million people that is cited by the UN-Habitat [132] and The World Bank [133,134]. Detailed statistical population forecasting was beyond the scope of our study. Still, our spatial analysis and policy review support the plausibility that by 2030, more Nairobi residents will rely on sanitation services that neither provide a good service from the individual perspective nor protect public health within the direct proximity nor are resilient to climate change impacts, particularly urban flooding (Table 5).

Our projections for the 2030 sewerage coverage fall considerably below NCWSC's targets. However, given the long-standing stagnation in sewer coverage rates and the challenge to meet existing water demand in Nairobi as well as documented challenges of water utilities in Kenya and elsewhere incentivising connections to existing sewers [117,135], we believe our projection of approximately 1.3 million additional people covered by sewerage by 2030 is rather optimistic. In fact, our model, based on proximity to sewer lines and income levels, may overestimate sewer connections (Fig 1).

We also acknowledge that our estimated prospects for expanding container-based sanitation services (CONTAINER regime) fall short of business expansion plans or the involved providers. We found no evidence of sustained funding that could drive a more significant expansion of road-based sanitation services within the current governance framework. The expansion of such services depends on household investments and external funding, and long-term sustainability will require public finance, which is not evident in current planning [36,64,65]. Currently, public finance for sanitation in Nairobi predominantly subsidises sewerage.

The impacts of climate change on all urban sanitation service regimes (Table 4) increasingly threaten service sustainability. Without targeted measures to enhance flood resilience, parts of Nairobi's population face escalating risks of sanitation system damage, malfunction, and temporary to permanent loss of access to adequate sanitation systems. Previous research shows that poor households often struggle to adapt their sanitation systems, reverting to unsafe practices in response to increased flooding [136–139]. Consequently, it is plausible to expect an increase in low-income residents relying on coping strategies or public toilets due to more frequent flooding. We chose not to quantitatively incorporate this factor to avoid suggesting false precision amidst cascading uncertainties. On the other hand, it is worth noting that expanding sewer systems and drainage in a coordinated manner (either through separated systems or well-designed combined sewers) could reduce urban flooding, provided that both systems are properly maintained and remain free from debris and solid waste blockages – a current challenge in Nairobi.

## Limitations

We acknowledge that our estimates on sanitation service configuration across Nairobi have considerable limitations as they are derived from a necessarily simplified allocation of dominant sanitation service regimes to sub-areas. These classifications

**Table 5. Comparision of the estimated distribution of population per sanitation service regime in Nairobi in 2022 (baseline) and 2030 rounded to 10,000s.**

|  | Baseline 2022 (see Table 2) | | Forecast for 2030 | | Delta | |
|---|---|---|---|---|---|---|
|  | Total | Proportion in % | Total | Proportion in % | Total | |
| COPING | 70,000 | 2 | 120,000 | 2 | +50,000 | – |
| PUBLIC TOILET | 70,000 | 2 | 120,000 | 2 | +50,000 | – |
| ST/NC | 450,000 | 10 | 730,000 | 12 | +280,000 | +2 |
| CONTAINER | 150,000 | 3 | 230,000 | 4 | +80,000 | +1 |
| ST/SEWER | 1,050,000 | 23 | 2,100,000 | 34 | +1,050,000 | +11 |
| HT/NC | 1,370,000 | 30 | 1,120,000 | 18 | −210,000 | −12 |
| HT/RT | 350,000 | 8 | 370,000 | 6 | 20,000 | −2 |
| HT/SEWER | 1,050,000 | 23 | 1,340,000 | 22 | +290,000 | −1 |
| Total | 4,560,000 | 100 | 6,130,000 | 100 | +1,010,000 | |

were informed by the prevailing technology and management arrangements identified within each area but inevitably obscure finer-grained overlaps, hybrid systems, and the coexistence of multiple service modalities. Consequently, both our spatial distribution and quantitative estimates should be viewed as indicative rather than definitive, acknowledging that each area reflects multiple and interacting service configurations shaped by broader political-economic processes [30].

We further recognise several limitations in our forecast of the sanitation regime configuration for 2030. Population projections are hugely complex. Elaborate statistical population forecasting was beyond the scope and intent of this study. The relatively short time series available for our forecast limits the reliability of the projections. In addition, our spatial model of the sanitation configurations consists of relatively small subareas, which further challenges the reliability of the associated raster data projections. As stated earlier, we provide a detailed description of our forecasting method and critical assessment of the limitations of our projection in the supplemental information (SI 4).

However, we remain confident that imprecisions in our quantifications do not substantially challenge the validity of our general findings, which are backed by our comprehensive analysis and review of the sanitation planning framework as well as the wider literature.

## Conclusion

In this paper, we have explored the rhetoric and realities of addressing sanitation inequities in Nairobi, particularly in the context of climate change adaptation. Nairobi's splintered sanitation sector is increasingly at risk due to the impacts of climate change. This vulnerability threatens to deepen the precarious and deeply inequitable sanitation service situation.

Our comprehensive review of Nairobi's legal, policy, and investment documents uncovers a critical gap between climate change and sanitation policy and planning. The evidence starkly highlights that the current policy and service provision are failing to deliver high-quality, equitable, and resilient sanitation services. Despite high-level government commitments (mainly on national government level) to ambitious targets concerning these aspects, a significant disconnect exists between the official pledges and the actual implementation on the ground. This gap is not only evident but is set to widen, deteriorating the sanitation experience for Nairobi's citizens as urbanisation, population growth and climate change impacts accelerate.

This analysis brings to light the concept of 'extraversion,' suggesting that the current political economy may foster a culture where development partners, government actors, and (international) non-governmental organizations collectively engage in practices that mainly improve the wording in well-intentioned policy and regulatory frameworks but avoid addressing the more challenging aspects of sanitation service provision and their societal root causes [88,89,140].

Despite acknowledging the necessity for service equity in Kenya's 'pro-poor' water and sanitation policy framework, there is a lack of effective measures to shift away from a conventional sewer-centric sanitation governance model. Instead, there is a glaring absence of a comprehensive citywide sanitation delivery framework, along with insufficient investment and operational subsidies for non-sewered sanitation service regimes. Consequently, sanitation planning in Nairobi seems destined to persist with a *'business-as-usual'* approach, prioritising sewerage development and neglecting the fragmented reality of its sanitation services as well as continuous (and often unplanned) urbanisation and increasing climate impacts on basic urban services.

Synthesising all findings from our analysis leads us to conclude that due to the inadequate integration of climate change and equity considerations in Nairobi's sanitation planning, it is highly plausible that by 2030, a substantial portion of residents will continue to depend on poor sanitation services. These services will likely fall short of providing good individual-level service quality and for safeguarding public health, particularly in the face of escalating climate threats.

The gaps exposed by our analysis are not limited to the local case of Nairobi. Splintered sanitation service regime configurations can be found in many major urban centres across Sub-Saharan Africa [31] and with modifications in low- and lower-middle income countries globally. Equally, 'fantasy planning' and extraversion are global phenomena. The congratulatory exchanges among government and development actors at policy forums and climate summits, while promoting policy advancements, do not address the core challenges in service provision.

Our analysis, therefore, underscores the urgent need for a fundamental shift in approach. Placing city governments at the centre of co-production processes for sanitation adaptation strategy [141] may offer effective paths towards adaptation plans that are more grounded in reality and go beyond fantasy planning. Further research in this area would be highly valuable. In addition, there must be an honest and robust discussion about providing high-quality, resilient services at scale beyond sewered areas, necessitating substantial government funding and support.

## Supporting information

**S1 Table. Summary of reviewed climate change policy and planning framework documents with focus on specific references to sanitation system adaptation and equity considerations.**
(PDF)

**S2 Table. Summary of reviewed sanitation sector policy and planning framework documents with focus on specific references to climate change adaptation and equity considerations.**
(PDF)

**S3 Table. Summary of reviewed sanitation investments in Nairobi county with focus on specific references to sanitation system adaptation and equity considerations.**
(PDF)

**S4 File. Description of population forecasting method using ArcGIS Pro time series forecasting tools.**
(PDF)

## Acknowlegments

The authors would like to thank Lais Dos Santos and Virginia Roaf for their constructive feedback on earlier versions of this manuscript. Our gratitude also goes to Ana Casas Garcia, who offered advice on population forecasting from a statistical viewpoint, and to Bart Hill for his guidance on the application of ArcGIS tools in population forecasting. Our thanks extend to Martin Ledant and Mutono Nyamai for their generosity in sharing shapefiles from their previous research on water inequalities in Nairobi. Additionally, we are deeply thankful for conversations with Anne Aol and Elizabeth Wambui, which offered invaluable insights into sanitation issues in Nairobi.

## Author contributions

**Conceptualization:** Leonie K. Hyde-Smith, Sarah Dickin.

**Data curation:** Leonie K. Hyde-Smith, Vanessa Reinfelder.

**Formal analysis:** Leonie K. Hyde-Smith.

**Funding acquisition:** Barbara Evans.

**Investigation:** Leonie K. Hyde-Smith.

**Methodology:** Leonie K. Hyde-Smith.

**Project administration:** Leonie K. Hyde-Smith.

**Software:** Vanessa Reinfelder.

**Supervision:** Daniel Ddiba, Sarah Dickin, Anna L. Mdee, Katy E. Roelich, Barbara Evans.

**Visualization:** Leonie K. Hyde-Smith.

**Writing – original draft:** Leonie K. Hyde-Smith.

**Writing – review & editing:** Leonie K. Hyde-Smith, Daniel Ddiba, Sarah Dickin, Domenic Kiogora, Anna L. Mdee, Katy E. Roelich, Barbara Evans.

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
