## [Decision Letter · Decision Letter 0]

29 Nov 2024

Dear Dr. Evans,

Thank you for submitting your manuscript to PLOS ONE. After careful consideration, we feel that it has merit but does not fully meet PLOS ONE’s publication criteria as it currently stands. Therefore, we invite you to submit a revised version of the manuscript that addresses the points raised during the review process.

We look forward to receiving your revised manuscript.

Kind regards,

Hope Onohuean, PhD

Academic Editor

PLOS ONE

**Journal Requirements:**

This work was supported by the UKRI Engineering and Physical Science Research Council (EPSRC) through a PhD studentship received by the first author (L.H.-S.) as part of the EPSRC Centre for Doctoral Training in Water and Waste Infrastructure and Services Engineered for Resilience (Water-WISER). EPSRC Grant number: EP/S022066/1

4. We note that Figures S1 - S4 and 3 - 4 in your submission contain map images which may be copyrighted. All PLOS content is published under the Creative Commons Attribution License (CC BY 4.0), which means that the manuscript, images, and Supporting Information files will be freely available online, and any third party is permitted to access, download, copy, distribute, and use these materials in any way, even commercially, with proper attribution. For these reasons, we cannot publish previously copyrighted maps or satellite images created using proprietary data, such as Google software (Google Maps, Street View, and Earth). For more information, see our copyright guidelines: http://journals.plos.org/plosone/s/licenses-and-copyright.

We require you to either present written permission from the copyright holder to publish these figures specifically under the CC BY 4.0 license, or remove the figures from your submission:

a. You may seek permission from the original copyright holder of Figures S1 - S4 and 3 - 4 to publish the content specifically under the CC BY 4.0 license.  

Reviewers' comments:

Reviewer's Responses to Questions

**Comments to the Author**

1. Is the manuscript technically sound, and do the data support the conclusions?

Reviewer #1: Yes

Reviewer #2: Partly

2. Has the statistical analysis been performed appropriately and rigorously?

Reviewer #1: Yes

Reviewer #2: No

3. Have the authors made all data underlying the findings in their manuscript fully available?

Reviewer #1: Yes

Reviewer #2: No

4. Is the manuscript presented in an intelligible fashion and written in standard English?

Reviewer #1: Yes

Reviewer #2: Yes

**Reviewer #1:**  Overall, the paper contributed to the body of knowledge on the intersections of climate change, urban sanitation, and service equity, with broader implications for policymakers, practitioners, and researchers working in this field. However, I have a few comments, questions, and observations:

Introduction

The introduction was well-written and effectively set the scene for the paper. However, starting the introduction with a numbered statement might not be the most appropriate approach. The introduction could further benefit from providing the readers with a clear articulation of this paper's unique contribution to the body of knowledge. The introduction should mention the broader significance and applicability of the research beyond the Nairobi case study, indicating the relevance to other urban centers facing similar challenges.

Methodology

The authors mention that the modeled changes in flood depth and extent between 2020, 2030, and 2050 showed only minor changes. This raises questions about the robustness of the flood risk assessment and whether it adequately captures the potential impacts of climate change on the sanitation system.

The authors do not provide detailed information on the specific data sources, methods, and assumptions used in the geospatial analysis to map the different types of sanitation systems. This lack of transparency may limit the replicability and evaluation of the approach.

Results

While the results highlight the persistent gaps between policy aspirations and practical implementation, especially in the critical areas of urban planning, climate resilience, and sanitation service delivery, the analysis could delve deeper into the underlying reasons for the disconnect between policy and implementation. The analysis could also draw parallels or contrasts with how the policy-implementation gaps manifest in other sectors.

Conclusion

The conclusion could include specific recommendations or a call to action for policymakers, government officials, development partners, and other stakeholders to address the identified gaps and challenges.

The conclusion could mention potential areas for future research that could build on the findings of this analysis.

**Reviewer #2:**  An worthwhile paper but needs to be refined with a clear purpose and aim. Reference to frameworks, methods and analytical results need more attention and focus. The conclusion raises at least 2 critical points that are impactful and need further explanation and support in the preceding discussion. See comments on attached file.

**Do you want your identity to be public for this peer review?** For information about this choice, including consent withdrawal, please see our Privacy Policy

Reviewer #1: **Yes: ** Olasunkanmi Habeeb Okunola

Reviewer #2: No

---

## [Author Response · Author response to Decision Letter 1]

15 Apr 2025

No Reviewer Section / Line Comment Response

1. Rev1 General Overall, the paper contributed to the body of knowledge on the intersections of climate change, urban sanitation, and service equity, with broader implications for policymakers, practitioners, and researchers working in this field. Thank you.

2. Rev1 Introduction The introduction was well-written and effectively set the scene for the paper. However, starting the introduction with a numbered statement might not be the most appropriate approach. The introduction could further benefit from providing the readers with a clear articulation of this paper's unique contribution to the body of knowledge. We have further clarified the aim/contribution of the paper as This sudy aims to examine the disparities between proclaimed and actual equity considerations at the intersection of urban sanitation planning and climate change adaptation in a city characterised by substanital sanitation inequalities in a resource constraint setting (p.4 line 10-104).

3. Rev1 Introduction The introduction should mention the broader significance and applicability of the research beyond the Nairobi case study, indicating the relevance to other urban centers facing similar challenges. See revised overall study aim (response above)

4. Rev1 Methodology The authors mention that the modeled changes in flood depth and extent between 2020, 2030, and 2050 showed only minor changes. This raises questions about the robustness of the flood risk assessment and whether it adequately captures the potential impacts of climate change on the sanitation system. We agree that this section needed clarification. Detailed flood modelling was beyond the scope of the study and therefore our ‘results’ lacked validity. We have revised the section accordingly.

5. Rev1 Methodology The authors do not provide detailed information on the specific data sources, methods, and assumptions used in the geospatial analysis to map the different types of sanitation systems. This lack of transparency may limit the replicability and evaluation of the approach. We provide the data sources of our shape files, satellite images and income specification.

• Mutono N, Wright J, Mutembei H, Thumbi SM. Spatio-temporal patterns of domestic water distribution, consumption and sufficiency: Neighbourhood inequalities in Nairobi, Kenya. Habitat International. 2022;119:102476.

• Ledant M. Access to Water in Nairobi: mapping inequalities beyond statistics. In: UN-Habitat Ga, editor. 2013.

• Google, cartographer Satellite imaginary of Nairobi2023.

We have added to the decription of validation (see response to comment 34

… to verify consistency and detect any changes in land use or income classification patterns. Key indicators, such as visible infrastructure developments, road networks, and urban expansion, were cross-checked to ensure the shapefiles matched current urban layouts.

Figure 1 decribes the mapping and categorisation process

6. Rev1 Results While the results highlight the persistent gaps between policy aspirations and practical implementation, especially in the critical areas of urban planning, climate resilience, and sanitation service delivery, the analysis could delve deeper into the underlying reasons for the disconnect between policy and implementation. Thank you we have added a summary paragraph addressing some the underlying issues: In summary, in Nairobi, the sanitation policy and planning framework mimic ‘recognition’ of sanitation inequities and climate threats whilst failing to translate their ambitious and performative statements into actions that would genuinely enhance the procedural or distributive equity within the ‘splintered sanitation sector’. Additionally, national policies in Kenya extravert ‘pro-poor’ inclusion within the water and sanitation sector as well as climate change mainstreaming. This is emblematic of a broader trend where development partners use aid conditions to incentivise governments to reproduce ‘best practice’ reforms of public sectors – such as the water sector – fostering the establishment of ostensibly ‘ideal’ institutions equipped with symbolic ‘good’ policy and institutional frameworks whilst lacking the requisite capability to fulfil their promises. These findings echo Narzetti and Marques {Narzetti, 2021 #6}analysis of reforms of Brazilian water and sanitation who describe the ineffectiveness of the de jure well designed ‘pro-poor’ and ‘inclusive’ water sector policy framework that de facto fails the 100 million Brazilians without wastewater collection services and largely excludes the peri-urban population.

7. Rev1 Results The analysis could also draw parallels or contrasts with how the policy-implementation gaps manifest in other sectors. We refer to flood resilience planning in Mumbai (p.20 lin 133 ff)

8. Rev1 Conclusion The conclusion could include specific recommendations or a call to action for policymakers, government officials, development partners, and other stakeholders to address the identified gaps and challenges. We have revised the last paragraph of the conclusion to address this comment: Our analysis, therefore, underscores the urgent need for a fundamental shift in approach. Placing city governments at the centre of co-production processes for sanitation adaptation strategy (141) may offer effective paths towards adaptation plans that are more grounded in reality and go beyond fantasy planning. Further research in this area would be highly valuable. In addition, there must be an honest and robust discussion about providing high-quality, resilient services at scale beyond sewered areas, necessitating substantial government funding and support.

9. Rev1 Conclusion The conclusion could mention potential areas for future research that could build on the findings of this analysis. See above

10. Rev2 General An worthwhile paper but needs to be refined with a clear purpose and aim. Reference to frameworks, methods and analytical results need more attention and focus. The conclusion raises at least 2 critical points that are impactful and need further explanation and support in the preceding discussion. See comments on attached file. Thank you. See our responses to the detailed comments below

11. Rev 2 Abstract Rewrite the abstract. Single paragraph. Include statements of method, main findings that can be briefly qualified, conclusion and recommendation (if any).

Abstract was revised

12. Rev2 Abstract Rewrite the abstract. Single paragraph. Include statements of method, main findings that can be briefly qualified, conclusion and recommendation (if any) Abstract was revised

13. Rev2 Abstract Qualify this statement: in the South this disjuncture between rhetoric and implementation is a common feature. Highlight a particular finding that qualifies this observation. The sentence that follows may be the qualification but it is unclear. Wording changed

14. Abstract Consider removing or qualifying the adjectives or pronouns that appear these final sentences that 'oversell' or exaggerate the conclusions. e.g. earnest discussion, fundamental, critical, substantial etc. Be clear about the conclusion and the recommendations if any. Wording changed

15. Introduction p.3 line58 Explain and consider using this to explain the title of the paper: what you intend to discuss in the using the words 'business as usual' and fantasy planning. Bring these ideas in earlier. Introduced terms in introduction

16. Introduction p.3 line62 Explain...there are many such frameworks with different emphases. Changed to … assessed through a justice lense… We give an explanation of the conceptualisation of climate adaptation justice in the following sentences

17. Introduction p.3 line69 f This is an important statement and should be the crux of the paragraph for developing the focus of the paper. Thank you. We have further clarified the aim of the paper see response to reviewer comment 2

18. Introduction p.3 line73 A good reason, but it limits the scope and generalisations that could be achieved from the paper. We have revised the section and think it is now clearer linked to the overall aim of the paper. Nairobi is used as an illustrative case for a city with severe sanitation inequalities while we also accept that there are limitations to generalisations from a distinct city case.

19. Introduction p.3 line75 Meaning obscure at this stage of the paper We have revised the section for more clarity

20. Introduction p.3 line77 Lots of jargon in this long sentence. Explain in more detail. We have revised the section for more clarity

21. Introduction p.4 line80 Referring to what? We have revised the section for more clarity

22. Introduction p.4 line80 Framework needs further explanation We have revised the section for more clarity

23. Introduction p.4 line81 Still too vague/confusing. Simplify. We have revised the section for more clarity

24. Introduction p.4 line82 What are these? Aligned with what? We have revised the section for more clarity

25. Introduction p.4 line86 Landscape? Sector? Regime? Lots of concepts that are left unexplained and result in confusion. We have revised the section for more clarity

26. Introduction p.4 line89 Explain. The lack of services, and flying toilets are not necessarily able to explain cause and effect relations, but are context specific in a social-economic setting. The reference may need more explanation. We have revised the section for more clarity

27. Introduction p.4 line94 ...and it certainly does, but is unable to reach the scale that this required. There are many innovative examples and efforts in Nairobi. Agreed – we have revised to acknowledge the initiatives : While there have been numerous applaudable and innovative sanitation efforts in Nairobi reflecting this commitment, these initiatives have struggled to achieve the scale necessary to effectively address the needs of marginalized populations.

28. Introduction p.4 line100 Before presenting the objectives, it will be important to clarify the overall aim and purpose of the paper (linked to the title?). Thank you. We have further clarified the aim of the paper see response to reviewer comment 2

29. Introduction p.5 line106 Needs to be explained in the preceding introduction. We have introduced the three common tenets of environmental / adaptation justice in the preceeding introduction and give a detailed interpretation of these in the context of sanitation adaptation in Table 2

30. Introduction p.5 line107f Perhaps too speculative? This is based on the results of a systematic review on climate impacts on sanitation failures as well as on climate scenarios that were modelled under the Hycristal project in collaboration with the UK Met Office. We therefore belive that our projections are grounded in robust data.

31. Introduction p.5 line109ff This objective needs further refinement. It appears to be a task, but it is unclear what is going to be achieved. Revised accordingly: Evaluate the adequacy of Nairobi’s policy and planning frameworks in addressing the additional pressures posed by climate change on sanitation systems. This involves assessing whether and how climate adaptation is incorporated into the city’s sanitation planning and investment framework, and conversely, whether sanitation considerations are integrated into Nairobi’s broader climate adaptation planning and investment framework. The aim is to determine whether these frameworks are fit-for-purpose to prevent increased vulnerabilities and inequalities resulting from potential climate hazard related sanitation system failures

32. Introduction p.5 line112f A duplication of objective #2? Explain. We changed the wording of the objective for more clarity. The difference to objective 2 is that objective assesses the likely impacts on the different sanitation service regimes on a theoretical level whereas objective 4 attempts to quantify the effects in terms of distributional equity

33. Method p.5 line117f What is this framework? What are indicators or metrics to be used? We describe the classifications and mapping process in Figure 1

34. Method p.5 line121 How was this achieved? Are there method statements? We added a further description … to verify consistency and detect any changes in land use or income classification patterns. Key indicators, such as visible infrastructure developments, road networks, and urban expansion, were cross-checked to ensure the shapefiles matched current urban layouts.

35. Method p.5 line122 What is this 'rapid' growth? Discussion missing. Added reference

36. Method p.5 line126 Will these sources, their validation and credibility be explained? Is the evidence robust? It is difficult to determine. We give the source of a recent base map and of the sanitation data (see also responses to comments 5 and 58

37. Method p.6 Figure 1 A poor map. Difficult to distinguish the categories. The table of explanation is blurred. A high resolution, colour map and a table of categories is recommended. Hi-res image included

38. Table 1 p6/7 Explain how Future 3 contradicts(?) 1 and 2. Future 3 is most extreme in terms of warming and also linked to erratic rainy seasons and prolonged dry spelts and droughts

39. p.7 line 156 Explain and why the selection was made See response to comment 41

40. p.7 line 158 unsubstantiated? See response to comment 41

41. p.8 line 161 Difficult to understand the value of this discussion section and purpose since it remains speculative and unsubstantiated. We agree that this section was confusing. The section has been removed. See response to reviewer comment 4

42. p.7 line 179 So, multiple sources were consulted. The method of extraction, indicators and framework were not discussed in any detail. This should be included in a discussion. Our thematic analysis is described in the following paragraph and Table 2 gives our analysis framework / evaluation critieria

43. p.8 line 186 Adapted? In what way? Does the discussion in the previous paragraph advance the adaptation of Wood and Roelich? In what way? Is there a connection? It reads like an illogical link. We based our framing of sanitation adaptation justice on Wood and Roelich’s exploration of energy justice and adapted it to the sanitation sector

44. p.8 line 179 which are what? Clarify Reformulated for clarity

45. p.9 line 198 (reference to S4 Annex) Not available to the reviewer. The process in developing these projections is unclear and cannot be verified from a short descriptive paragraph. The forecasting method is described in detail in the Supplementary Material

46. p.9 line201 Redundant sentence We believe that this sentence has value as sanitation is still understood as mainly infrastructure by many

47. p. 9 line 202 Still underexplained since there are many frameworks and concepts of so called socio-technical regimes. We moved this section to the introduction and revised the corresponding section

48. p.9 line 204 Sure, but what is the position and conceptual understanding of the authors and how does this connect to the aim, purpose and objectives of the paper? See above

49. p.9 line 211 Sure, but this is an uncritical approach to context and success of informal and decentralised systems in a 'chaotic' densely settled peri urban city. Its much more complex and that centralised systems are not the only 'solution'. We fully agree on the point made about centralised systems not being the only solution. This is also reflected in our assessment of distributional equity shown in figure 2 which acknowledges the role of full service chain decentralised solutions as part of an adequate sanitation service provision. Unfortunately we understand from comment 51 that the reviewer was not able to access the figure.

50. p.10 line 221 Sure, shows some recognition of the scope and limitations of the study and thinking. Good to read this? OK

51. p.10 line 223 (Figure 2) Not available to reviewer. All links in Annexure were not functional OK

52. p.10 line 227 Who are these residents? Is there empirical evidence and a method? Reference added

53. p.10 line 232 Evidence or a reference? Reference added

54. P.11 line 264 The scale changes again from the discussio

---

## [Decision Letter · Decision Letter 1]

27 Jun 2025

Dear Dr. Evans,

Thank you for submitting your manuscript to PLOS ONE. After careful consideration, we feel that it has merit but does not fully meet PLOS ONE’s publication criteria as it currently stands. Therefore, we invite you to submit a revised version of the manuscript that addresses the points raised during the review process.

**Dear authors, I totally agree with reviewer #4 minor correction. Please modify the manuscript to accommodate the minor adjustments to make this paper ready for publication.**

We look forward to receiving your revised manuscript.

Kind regards,

Hope Onohuean, PhD

Academic Editor

PLOS ONE

**Journal Requirements:**

Reviewers' comments:

Reviewer's Responses to Questions

**Comments to the Author**

Reviewer #3: All comments have been addressed

Reviewer #4: All comments have been addressed

2. Is the manuscript technically sound, and do the data support the conclusions?

Reviewer #3: Yes

Reviewer #4: Yes

3. Has the statistical analysis been performed appropriately and rigorously?

Reviewer #3: N/A

Reviewer #4: Yes

4. Have the authors made all data underlying the findings in their manuscript fully available?

Reviewer #3: Yes

Reviewer #4: Yes

5. Is the manuscript presented in an intelligible fashion and written in standard English?

Reviewer #3: Yes

Reviewer #4: Yes

**Reviewer #3: ** This manuscript spotlights challenges of sanitation adaptation in a climate-vulnerable urban context. It combines political realism with technical shifts. It is a good contribution to both urban climate governance literature and sustainable development discourse. There has been an enhanced clarity and structure, I have no additional comments this time.

**Reviewer #4:**  The authors have modified the manuscript based on the previous suggestions but there are some minor adjustments to make this paper ready for publication. Here are some:

• Kindly ensure any framework mentioned is named and explicitly explained to convey the idea to potential reader. Explain why a particular framework is selected over the others.

• While multiple data sources are used, on-ground validation appears limited

• The transition is abrupt between the sections and technical terms need to be adequately explained.

• Is there any on-ground validation of the satellite imagery/ shapefiles used in the research?

o Strengthen the justification for adapting Wood & Roelich (Table 2)

o Detail the criteria used to assign a "dominant" regime to each area, especially where heterogeneity exists. How were overlaps or ambiguities resolved?

o Provide more specifics on "validation and refinement" using stakeholder input and studies. What discrepancies were found? How were they reconciled? A short appendix table summarizing validation sources and key adjustments would be valuable.

o Some few grammatical errors to be corrected include:

Line 39: “It calls for an robust and honest…..” should be changed to “It calls for a robust and honest….”

Line 89-90: “on the sanitation sector of Nairobi and Kenya …” should be changed to “on the sanitation sector of Nairobi, Kenya.

**Do you want your identity to be public for this peer review?** For information about this choice, including consent withdrawal, please see our Privacy Policy

Reviewer #3: No

Reviewer #4: No

---

## [Author Response · Author response to Decision Letter 2]

12 Nov 2025

Reviewer #4: The authors have modified the manuscript based on the previous suggestions but there are some minor adjustments to make this paper ready for publication. Here are some:

• Comment 1: Kindly ensure any framework mentioned is named and explicitly explained to convey the idea to potential reader. Explain why a particular framework is selected over the others.

Response: We understand the reviewer’s concern and have revised the text to explicitly name and explain the socio-technical regime framework. We have also clarified why this framework was selected, citing previous studies by Welie et al. (31) and Mdee et al. (30), who have already applied it to analyse Nairobi’s sanitation sector. Their work provides a strong conceptual foundation for our study (see lines 80 ff).

• Comment 2: While multiple data sources are used, on-ground validation appears limited

Response: We agree and explicitly acknowledge this limitation in the manuscript. Our analysis adopts a citywide approach, as literature on sanitation adaptation at this scale remains limited, particularly for cities with heterogeneous sanitation systems. Please also refer to our response to comment 4.

Comment 3: The transition is abrupt between the sections and technical terms need to be adequately explained.

Response: We suspect the reviewer might not have accessed figure 2 in which we define the sanitation regimes

Comment 4: Is there any on-ground validation of the satellite imagery/ shapefiles used in the research?

Response: Comprehensive on-ground validation was not feasible given the citywide scale of analysis. However, we validated our shapefiles by cross-referencing them with recent satellite imagery and an updated basemap (see lines 124 ff) In addition we consulted expert opinions. We also provide an example of local on-ground validation using for the Mukuru area (lines 255–259).

• Comment 5: Strengthen the justification for adapting Wood & Roelich (Table 2)

Response: We strengthened the justification given in lines 179ff to As no coherent definition for ‘just’ or ‘equitable’ climate adaptation of urban sanitation currently exists, we drew upon environmental and energy justice scholarship (22). We adapted the energy justice framework proposed by Wood and Roelich (22) to the urban sanitation context by translating its justice dimensions into evaluative aspects of equitable sanitation adaptation (Table 2).

• Comment 6: Detail the criteria used to assign a "dominant" regime to each area, especially where heterogeneity exists. How were overlaps or ambiguities resolved?

esponse: The process is described in Figure 1 We clearly state in the limitations that we acknowledge the impreciseness of our estimates. We have further sharpened this section (line 337ff): We acknowledge that our estimates on sanitation service configuration across Nairobi have considerable limitations as they are derived from a necessarily simplified allocation of dominant sanitation service regimes to sub-areas. These classifications were informed by the prevailing technology and management arrangements identified within each area but inevitably obscure finer-grained overlaps, hybrid systems, and the coexistence of multiple service modalities. Consequently, both our spatial distribution and quantitative estimates should be viewed as indicative rather than definitive, acknowledging that each area reflects multiple and interacting service configurations shaped by broader political-economic processes (30).

• Comment 7: Provide more specifics on "validation and refinement" using stakeholder input and studies. What discrepancies were found? How were they reconciled? A short appendix table summarizing validation sources and key adjustments would be valuable.

Response: We provide a description of our iterative approach to the sanitation regime classification in figure 1. We also provide a description of our steps to estimate the population in Annex 4 (SI). More detail is given in the text lines 122 – 145 as well as in the limitation section lines 341-347.

• Comment 8: Some few grammatical errors to be corrected include:

o Comment 8a: Line 39: “It calls for an robust and honest…..” should be changed to “It calls for a robust and honest….”

o Response: Changed as suggested

o Comment 8a: Line 89-90: “on the sanitation sector of Nairobi and Kenya …” should be changed to “on the sanitation sector of Nairobi, Kenya.

o Response: This is correct as it is written, we refer to both the studies specific to Nairobi and also more broader studies about the sanitation sector in Nairobi

---

## [Editor Report · Decision Letter 2]

4 Dec 2025

Business-as-usual and fantasy planning – an analysis of equity within climate adaptation planning for sanitation in Nairobi

PONE-D-24-10746R2

Dear Dr. Evans,

We’re pleased to inform you that your manuscript has been judged scientifically suitable for publication and will be formally accepted for publication once it meets all outstanding technical requirements.

Kind regards,

Hope Onohuean, PhD

Academic Editor

PLOS One
---

## [Editor Report · Acceptance letter]

PONE-D-24-10746R2

PLOS One

Dear Dr. Evans,

I'm pleased to inform you that your manuscript has been deemed suitable for publication in PLOS One. Congratulations! Your manuscript is now being handed over to our production team.

Kind regards,

on behalf of

Dr. Hope Onohuean

Academic Editor

PLOS One